# Correlates of Inaccuracy in Reporting of Energy Intake Among Persons with Multiple Sclerosis

**DOI:** 10.3390/nu17030438

**Published:** 2025-01-25

**Authors:** Stephanie L. Silveira, Brenda Jeng, Barbara A. Gower, Gary R. Cutter, Robert W. Motl

**Affiliations:** 1Department of Physical Therapy, University of Alabama at Birmingham, 3810 Ridgeway Drive, Birmingham, AL 35209, USArobmotl@uic.edu (R.W.M.); 2Department of Management, Policy, and Community Health, University of Texas Health Science Center at Houston School of Public Health, 1200 Pressler Street, Houston, TX 77493, USA; 3Department of Kinesiology and Nutrition, University of Illinois Chicago, 1919 W. Taylor Street, Chicago, IL 60612, USA; 4Department of Nutrition Sciences, University of Alabama at Birmingham, 1675 University Blvd, Birmingham, AL 35233, USA; bgower@uab.edu; 5Department of Biostatistics, University of Alabama at Birmingham, 1675 University Blvd, Birmingham, AL 35233, USA; cutterg@uab.edu

**Keywords:** energy intake, cognition, physical activity, hydration, doubly labeled water

## Abstract

Background/Objectives: Persons with multiple sclerosis (MS) are interested in diet as a second-line approach for disease management. This study examined potential variables that correlate with inaccuracy of self-reported energy intake (EI) in adults with MS. Methods: Twenty-eight participants completed two assessment appointments within a 14-day period that included a standard doubly labeled water (DLW) protocol for estimating total energy expenditure (TEE). The participants reported their EI using the Automated Self-Administered 24 h (ASA24) Dietary Assessment Tool. The primary variables of interest for explaining the discrepancy between TEE and ASA24 EI (i.e., inaccuracy) included cognition (processing speed, visuospatial memory, and verbal memory), hydration status (total body water), and device-measured physical activity. Pearson’s correlations assessed the association between absolute and percent inaccuracy in reporting of EI with outcomes of interest, followed by linear regression analyses for identifying independent correlates. Results: California Verbal Learning Test—Second Edition (CVLT-II) z-scores and light physical activity (LPA) were significantly associated with mean absolute difference in EI (*r* = –0.53 and *r* = 0.46, respectively). CVLT-II *z*-scores and LPA were the only variables significantly associated with mean percent difference in EI (*r* = –0.48 and *r* = 0.42, respectively). The regression analyses indicated that both CVLT-II and LPA significantly explained variance in mean absolute difference in EI, and only CVLT-II explained variance for percent difference in EI. Conclusions: The results from this study indicate that verbal learning and memory and LPA are associated with inaccuracy of self-reported EI in adults with MS. This may guide timely research identifying appropriate protocols for assessment of diet in MS.

## 1. Introduction

Multiple sclerosis (MS) is a chronic, neurodegenerative disease of the central nervous system (CNS) [1]. The CNS damage manifests as a broad set of neurological symptoms, such as mobility impairment, cognitive impairment, fatigue, and mood disturbances [1]. The current estimated prevalence of MS in the United States is one million cases [2], and the most common treatments are disease-modifying drug therapies that prevent relapses, reduce disability progression, and modify the disease course [1]. There are currently limited second-line approaches available for MS disease management (e.g., lifestyle behaviors), although persons with MS experience significant barriers, such as patient dissatisfaction and lack of insurance coverage, to accessing traditional medical therapies [3].

Diet represents the top lifestyle topic of interest for managing disease and symptomatic outcomes among persons with MS [4]. Indeed, persons with MS believe that debilitating symptoms can be effectively managed through diet [5]. Research further indicates that persons with MS, particularly those recently diagnosed with the disease, often self-experiment with diet change as an MS management strategy [6,7]. Foundational research examining diet among persons with MS has utilized various self-report measures, including food frequency questionnaires, food diaries, diet screening questionnaires, and brief dietary assessment instruments [8,9,10,11]. The study of dietary changes for managing MS requires high-quality research designs, including validated measures of diet and energy intake.

One recent study examined the validity of measures of diet among persons with MS by comparing doubly labeled water (DLW) and self-reported energy intake using the Automated Self-Administered 24 h (ASA24) Dietary Assessment Tool in persons with MS [12]. That study reported no significant correlation between DLW total energy expenditure (TEE) and ASA24 mean kcal/day, and further estimated an average 25% underreporting of energy intake in MS [12]. The next step in this line of research involves examining variables that may explain inaccuracy in reporting of energy intake based on debilitating symptoms, particularly cognitive dysfunction (i.e., memory, recognition, and recall that are needed for common diet assessments) [5], and lifestyle behaviors that influence measurement of energy intake and expenditure.

There may be MS-specific variables associated with inaccuracy in reporting energy intake. For example, an estimated 45–65% of persons with MS experience significant cognitive dysfunction [13]. Self-report measures of energy intake require recall and accuracy, which may be significantly more challenging and less valid in persons with MS based on the presence of memory dysfunction. The 25% underreporting of energy intake reported in previous research may be associated with cognitive dysfunction that is commonly reported in the MS population [12]. One additional MS-specific consideration is hydration status, given persons with MS report significant bowel and bladder dysfunction that can influence water consumption and, in turn, estimates of TEE using biospecimen analyses that utilize urine [14]. The examination of MS-specific influences on common diet measures is key to establishing validity in this unique population and may provide guidance regarding key variables that influence dietary measurement for informing future research and clinical practice.

Another factor in the TEE equation that may account for inaccuracy in reporting of energy intake among persons with MS is physical activity. Physical activity is of particular interest in the current study given significant differences in physical activity between persons with MS and healthy controls wherein persons with MS are significantly less active [15]. Physical activity is the most variable component when estimating TEE in standard equations because of significant inter-individual variation [16]. Therefore, when examining inaccuracy in reporting of energy intake compared to DLW estimation, physical activity level is a primary variable of interest.

The current study focused on examining potential variables that contribute to inter-individual variation in accuracy of reporting of energy intake in a sample of persons with MS. Such an inquiry is crucial for identifying factors of particular relevance among people with MS that need to be considered when measuring diet in this population. Accurate measurement of diet is the foundation for the clinical management of MS using diet, given a metric must be established to assess compliance with dietary interventions. The primary variables of interest included cognition (processing speed, visuospatial memory, and verbal learning and memory), hydration status (total body water), and physical activity. The research team hypothesized that poor cognition would be the primary contributor to inaccurate reporting of energy intake in this sample, given that self-report of dietary intake relies on information processing and recall that are often impacted by MS.

## 2. Materials and Methods

### 2.1. Participants

This paper involved a secondary analysis of data from a cross-sectional study validating energy intake assessment in persons with MS compared with healthy controls matched for age and biological sex on a 2:1 basis conducted between October 2019 and September 2020 [12]; healthy control participants were not included in this secondary analysis of data as we focused on correlates of inaccuracy of direct relevance for MS. The primary study sample included 30 participants who completed telephone screening and met the following inclusion criteria: (a) diagnosis of MS; (b) relapse-free for the past 30 days; (c) willingness to complete two study visits including questionnaires, 6 ASA24 diet recalls, and a DLW protocol; (d) age between 18 and 55 years; and (e) access to Internet and e-mail for ASA24 completion. The exclusion criteria included self-report weight loss of 10 or more pounds over the past three months, and two participants were excluded in this study due to invalid DLW assessments. This yielded the final sample of 28 participants with MS (Figure 1).

### 2.2. Procedure

The University of Alabama at Birmingham Institutional Review Board approved the study protocol, and all participants provided written informed consent. The participants completed a fasted baseline assessment visit at the University research laboratory, which included a standard DLW protocol, bioelectrical impedance analysis, a neuropsychological assessment battery, a battery of questionnaires (i.e., demographics and clinical characteristics and physical activity), and an ASA24 diet recall. Following the baseline assessments, the participants were asked to complete a 7-day physical activity measurement protocol using an accelerometer. The participants then returned for another assessment visit at Day 14, which included a urine sample for DLW estimation and an ASA24 diet recall. The participants were prompted to complete two additional ASA24 diet recalls on two random days following each assessment with a total goal for completion of 6 ASA24 diet recalls. The participants were provided with compensation upon completion of each assessment visit.

### 2.3. Measures

#### 2.3.1. Total Energy Expenditure

A DLW method was used to estimate TEE, reported as kcal/day [17]. A standard protocol was applied that is thoroughly described elsewhere [12]. Briefly, the participants completed a fasted baseline urine sample prior to oral DLW administration. The DLW solutions used were Cortecnet Oxygen-18 (H_2_^18^O) Isotope Enrichment ≥10% and Deuterium (D^2^O) 99.8% atom D, and the dose administered was calculated based on self-reported body weight during telephone screening: ^18^O:D_2_O is 1 g:0.08 g by weight, wherein the participants were dosed 1 g total solution per kg = 0.926 g/kg of water with 10% ^18^O atoms and 0.074 g/kg of water with 99.8% ^2^H atoms [18]. Dosed urine samples were collected at three and four hours, with 4 h samples used for all TEE calculations. The participants were scheduled to complete a final urine sample at Day 14 assessment with two participants requiring alternate days (Day 12 and 13) due to scheduling conflicts. Samples were analyzed in duplicate for H_2_^18^O and ^2^H_2_O enrichments by isotope-ratio mass spectrometry (IRMS) on a Thermo Scientific Delta V Advantage IRMS with Gas Bench. Turnover rates and zero-time extrapolated dilution spaces of H^2^_18_O and ^2^H_2_O were calculated from the slope and intercept of the semi-logarithmic plot of isotope enrichment in urine, versus time after dosing, using the Coward equation [19]. CO_2_ production rates and TEE were calculated based on the recent updated equations of Speakman et al. [20].

#### 2.3.2. Self-Reported Energy Intake

Energy intake was self-reported using the Automated Self-Administered 24 h (ASA24) Dietary Assessment Tool (version 2018) developed by the National Cancer Institute [21]. The ASA24 is a self-administered online tool that includes a validated, multi-pass 24 h diet recall protocol [22]. Briefly, the ASA24 website is equipped with a series of steps to capture all food and beverages consumed during the previous 24 h period. The overarching aim in the study was for the participants to complete three ASA24 diet recalls within a week of each assessment appointment (i.e., 6 ASA24 diet recalls in total). The participants completed their first ASA24 during their baseline assessment appointment and 4th recall during their Day 14 assessment appointment. The participants were prompted via e-mail/text to complete two recalls in the week following each appointment on two random non-consecutive days. The ASA24 protocol was aligned with best practices outlined in previous research wherein participants were prompted to complete three recalls during a one-week period on two random weekdays and 1 random weekend day [23]. Any days with a reported energy intake below 500 calories were excluded (*n* = 6 ASA24 diet recalls excluded among 5 participants) [23,24]. Among this sample, 22 participants completed six ASA24 diet recalls, 3 participants completed five ASA24 diet recalls, and 3 participants completed four ASA24 diet recalls.

#### 2.3.3. Total Body Water

Bioelectrical impedance analysis (BIA) was used to measure total body water (TBW) during the fasted baseline assessments using a Seca^®^ mBCA 514 Medical Body Composition Analyzer (Hamburg, Germany). TBW was estimated using the device’s patented equation. Body mass index (BMI) was calculated using height and weight from measurement.

#### 2.3.4. Cognitive Assessments

The Brief International Cognitive Assessment for Multiple Sclerosis (BICAMS) was utilized to assess cognition [25]. BICAMS is considered the international standard brief battery of tests for cognitive evaluation in MS and was compiled with the intent for easy implementation in clinical settings and comprehensive assessment of the most prevalent domains of cognitive impairment in MS [25]. All tests were administered using a standardized script and protocol regarding practice and number of trials. The oral response Symbol Digit Modalities Test (SDMT) measured cognitive processing speed in persons with MS [26,27]. During the SDMT, participants are provided with a sheet of paper with a standard series of symbols and requested to say aloud the digit that pairs with each symbol according to a key that is displayed at the top of the page during one 90 s trial. The SDMT score represents the total number of symbols accurately paired with their digit with a total range of possible scores from 0 to 110. The SDMT is recognized as a particularly sensitive, reliable, and valid measure of processing speed in MS, given its strong correlation with MRI-measured central atrophy (*r* = 0.70 or 0.71) [27]. The participants then completed five trials of the California Verbal Learning Test—Second Edition (CVLT-II), a valid measure of verbal learning and memory in persons with MS [28,29]. A member of the research team read a list of 16 words aloud, and then the participants were instructed to immediately recall as many words as possible in any order. The total CVLT-II composite score is the total number of words recalled across the 5 consecutive trials with a total possible score of 80. The CVLT-II has been validated in persons with MS by examining correlation with ventricular brain fraction (*r* = −0.49) [30]. Lastly, the participants completed 3 trials of the Brief Visuospatial Memory Test—Revised (BVMT-R), a valid measure of visuospatial learning and memory in persons with MS [31]. During each trial, a member of the research team displayed six figures for 10 s, and after the display was removed, the participants were asked to draw each figure precisely and in the correct location on the page. Each trial is scored separately with a possible score of 0, 1, or 2 for each figure based on the precision and location of the figures, yielding a total possible score of 36. Notably, the BVMT-R tests a unique domain of learning and memory that relies on visual acuity, which is commonly impacted by MS disease [32]. The validity of the BVMT-R has been established by examining correlation with thalamic volume in persons with MS (*r* = 0.51) [33].

#### 2.3.5. Physical Activity

Physical activity was device-measured using an ActiGraph GT3X+ accelerometer. A standard 7-day protocol was utilized wherein the participants were requested to wear the accelerometer on an elastic belt on their non-dominant hip during waking hours for seven days. Accelerometer data were downloaded using the ActiLife software [34], and the data were processed into 60 s epochs with low frequency extension. Moderate-to-vigorous physical activity (MVPA) was calculated based on a MS-specific cut-point of 1745 counts/min [35]. Count values below 100 counts/min were considered sedentary behavior, and count valuea between 100 and 1744 counts/min were classified as light physical activity (LPA). Days were considered valid if wear time was ≥600 min [36].

#### 2.3.6. Participant Demographics and Clinical Characteristics

The participants completed a self-report questionnaire that included biological sex, marital status, age, employment status, race and ethnicity, income, and level of education. The self-reported clinical characteristics included MS clinical course, years since MS diagnosis, and the Patient-Determined Disease Steps [37].

### 2.4. Data Analyses

All data analyses were conducted using SPSS Statistics [38]. Descriptive statistics were reported as mean ± standard deviation unless otherwise noted (e.g., median and interquartile range [IQR] or number and percentage) based on the type of variable (i.e., continuous vs. ordered-categorical) and assessments of normality per variable. Absolute difference values for inaccuracy in reporting of energy intake were calculated by subtracting TEE from mean ASA24, and the percent difference was then calculated by taking the absolute difference value and dividing it by TEE and multiplying by 100. Each participant’s absolute and percent difference values were calculated, with the results reflecting the overall mean of each participant’s difference values. All cognitive assessment scores were converted to z-scores for data analyses to account for age, sex, and years of education using a standard protocol [39]. Pearson’s correlations (r) were utilized to assess the association between absolute and percent inaccuracy in reporting of energy intake with the outcomes of interest. Values for correlation coefficients of 0.1, 0.3, and 0.5 were interpreted as weak, moderate, and strong, respectively [40]. Linear regression analyses were conducted using simultaneous entry of variables whereby the significant variables from the correlation analyses were included as independent correlates of absolute and percent inaccuracy in reporting of energy intake.

## 3. Results

### 3.1. Participants

The participant demographics and clinical characteristics are presented in Table 1. The mean ± SD for the outcomes of interest included in the subsequent correlation analyses is presented in Table 2. The mean DLW TEE as an estimation of energy intake was 1894 ± 319, and the mean ASA24 reported energy intake across all assessments (i.e., up to 6 days’ total per participant) was 1601 ± 396. The ASA24 underestimated energy intake by approximately 600 ± 529 kcal and/or 25 ± 21% compared to DLW. The mean SDMT *z*-score was −0.9 ± 1.1, the BVMT-R *z*-score was −0.5 ± 1.2, and the CLVT-II *z*-score was −1.0 ± 1.0.

### 3.2. Bivariate Correlations Between Inaccuracy in Reporting of Energy Intake and Outcomes of Interest

The correlation coefficients for primary variables associated with differences in energy expenditure are provided in Table 3. The CVLT-II *z*-scores and LPA were the only variables significantly associated with mean absolute difference in energy intake (*r* = –0.53 and *r* = 0.46, respectively). The CVLT-II *z*-scores and LPA were significantly associated with mean percent difference in energy intake (*r* = –0.48 and *r* = 0.42, respectively).

### 3.3. Linear Regression Analyses

Two linear regression analyses were conducted for examining the independent contributions of variables (i.e., CVLT-II and LPA scores) for explaining variance in the absolute and percent difference in energy intake. Regarding absolute difference in energy intake, the results of the regression indicated that both CVLT-II scores and LPA significantly explained the variance [F(2, 25) = 8.41, *p* = 0.001, R^2^ = 0.40] (Table 4). Regarding percent difference in energy intake, the results of the regression indicated that only CVLT-II scores significantly explained the variance, [F(2, 25) = 6.13, *p* = 0.007, R^2^ = 0.33]. LPA did not significantly contribute to the model.

## 4. Discussion

This is the first study that examined MS-specific factors associated with inaccuracy of self-reported energy intake, and it may guide timely research identifying valid tools for measurement of diet in MS. Further, this study is the first to examine measures of clinical relevance in MS that can be assessed in community settings for informing appropriate dietary intervention tools and approaches. The results indicated that verbal learning and memory was associated with inaccuracy in reporting of energy intake (i.e., both absolute and percent difference) in persons with MS. Additionally, device-measured LPA was associated with the absolute difference in energy intake, which is the most commonly employed metric for assessing accuracy of energy intake reporting. The analyses indicated that device-measured MVPA, TBW, cognitive processing speed, and visuospatial learning and memory did not significantly account for inaccuracy in reporting of energy intake in this sample. Overall, this suggests that specific components of cognitive impairment, namely verbal learning and memory, as well as engagement in lower-intensity physical activity (i.e., LPA), were the primary variables related to error in self-reporting of diet in persons with MS.

The results from this study indicated that lower neuropsychological test scores were associated with greater inaccuracy in reporting of energy intake; however, the follow-up analyses indicated that only verbal learning and memory significantly contributed to discrepancies between measures of diet. Measures of memory and recall were included as an outcome of interest in the current study due to previous research highlighting cognitive decline and dysfunction among older adults as a potential factor related to the validity of recall-based self-report measures of diet [41]. The current study provides strong support for the potential of verbal learning and memory as a significant variable associated with error in reporting of energy intake and further inquiry regarding cognitive processing speed. Further research may assess the use of techniques that may aid in memory and recall, such as documenting all food intake using a journal or photographs during the data collection period.

Physical activity is a key component of the TEE equation [16]; however, only device-measured LPA was associated with absolute differences in inaccuracy in reporting of energy intake in the current study. Specifically, engagement in more minutes of LPA was associated with higher absolute difference between self-reported energy intake and biologically measured TEE. It is crucial to note that associations between LPA and cognitive functioning have not been consistent in persons with MS [42], which provides an important area to investigate in future research given both variables significantly contributed to underreporting energy intake. It is surprising that MVPA was not associated with differences in energy intake or TEE in this sample; however, it is possible that there was a floor effect in the current sample wherein rates of physical of activity were low and did not vary in a large magnitude that would be necessary to illuminate significant differences. Previous research in healthy populations indicates that individuals who are more active may be less accurate in reporting dietary intake [43]. The current findings align with other research in healthy populations highlighting engagement in light leisure time activity being related to a larger discrepancy between self-reported energy intake and TEE compared to vigorous activity [44]. Further inquiry is needed to help illuminate whether LPA is compensated for with increased dietary intake among weight-stable adults with MS; such compensation may have led to the pattern of error in this sample. Taken together, physical activity is an important factor to consider in future studies examining TEE in persons with MS. Associations between energy intake and energy expenditure related to physical activity may inform specific factors to consider when determining whether results from healthy populations generalize to persons with MS.

TBW was not a significant variable associated with inaccuracy in reporting of energy intake in MS. The inclusion of TBW was grounded in the hypothesis that hydration status of persons with MS is lower than the general population, given chronic self-imposed dehydration due to bowel and bladder dysfunction [14]. The lack of association between TBW and inaccuracy in reporting of energy intake provides a novel contribution to the literature wherein this preliminary evidence may reduce concerns that TBW is a source of invalidity in estimation of energy intake using DLW in MS. However, this study was the first application of DLW in MS, and future replication studies are warranted using alternative biospecimen analyses for measurement of DLW, such as blood or saliva.

This study included several strengths, such as the use of DLW as the gold standard measure of energy intake and recruitment of a sample that included a large portion of individuals who identified as Non-Hispanic Black or African American. However, one of the primary limitations of the current study is the small sample size from one geographic area given funds were not sufficient for a larger sample. Additionally, this study included only four participants who identified as male, limiting the generalizability to the broader MS population. Future studies are needed that include larger, more representative samples varying in both cognitive functioning and physical activity and across a range of geographic locations. These studies should focus on differences due to race/ethnicity, as there is a growing body of literature examining the impact of social determinants of health on MS symptoms and outcomes. An additional source of error to consider is random error, such as variability in diet. For example, the participants in the study included individuals enrolled both prior to and after COVID-19 restrictions that may have had a significant impact on energy intake; however, we assert that changes in energy intake likely would not influence the reporting of diet beyond standard confounding associated with social desirability [45,46].

## 5. Conclusions

The current study provides a novel contribution to the nutrition literature regarding the assessment of energy intake in persons with MS. Verbal learning and memory was the primary variable associated with inaccuracy in reporting of energy intake, which may be a significant variable to consider in other populations with cognitive dysfunction. Physical activity, particularly LPA, was associated with absolute differences in energy intake that should be further examined as a key variable in the TEE equation. This line of research has established that general population protocols for reporting energy intake are not sufficient in persons with MS, with the current study focused on identifying factors of interest to guide future research. Accurate measurement of diet and factors that contribute to disproportionate inaccuracy is foundational for the application of second-line therapies among MS clinicians. Overall, there is a prime opportunity for innovative research examining appropriate tools and procedures for measuring diet in MS that are critically needed for examining compliance with dietary interventions.

## Figures and Tables

**Figure 1 nutrients-17-00438-f001:**
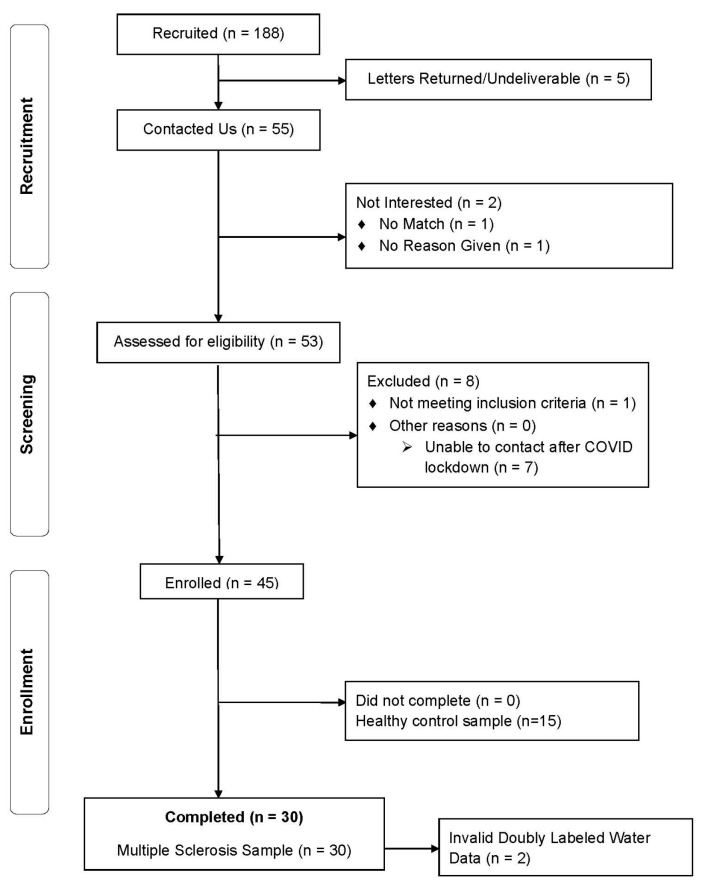
Flow diagram of recruitment and enrollment of participants for study examining the validity of energy intake reporting among persons with multiple sclerosis.

**Table 1 nutrients-17-00438-t001:** Demographic and clinical characteristics of 28 participants with multiple sclerosis enrolled in a cross-sectional study examining accuracy of self-reported energy intake.

Variable, Units	Mean ± SD
**Age**, years	39.8 ± 9.6
**MS ^a^ Duration**, years	10.0 ± 6.5
**PDDS ^b^,** rating	**Median (IQR)** 0 (2)
**Sex** Female Male	**N (%)** 24 (85.7) 4 (14.3)
**Marital Status** Married Single/Divorced/Widowed	14 (50.0) 14 (50.0)
**Employed** Yes No	19 (67.9) 9 (32.1)
**Race and Ethnicity** Non-Hispanic White Non-Hispanic Black or African American Hispanic or Latino	11 (39.3) 16 (57.1) 1 (3.6)
**Education** High School-Some College College Graduate or More	9 (32.1) 19 (67.9)
**Annual Household Income** $50,000 or Less Greater than $50,000	11 (40.7) 16 (59.3)
**MS Clinical Course** Relapsing Remitting MS Progressive	26 (92.9) 2 (7.1)

Note. ^a^ MS: Multiple Sclerosis; ^b^ PDDS: Patient-Determined Disease Steps measure of disability severity (range 0–8).

**Table 2 nutrients-17-00438-t002:** Descriptive statistics for outcomes of interest (i.e., difference in self-reported energy intake, hydration status, cognition, and device-measured physical activity) examined in correlation analysis among a sample of 28 persons with multiple sclerosis.

Variable, Units	Mean ± SD
**Absolute Difference EI ^a^,** kcals	585 ± 529
**Percent Difference EI,** %	24.6 ± 21.3
**TBW ^b^,** kg	35.6 ± 5.2
**TBW%,** %	45.1 ± 5.5
**SDMT ^c^,** z-score	−0.90 ± 1.06
**CLVT-II ^d^,** z-score	−0.97 ± 1.02
**BVMT-R ^e^,** z-score	−0.48 ± 1.18
**LPA ^f^,** Minutes/day	296.0 ± 78.7
**MVPA ^g^,** Minutes/day	23.1 ± 18.3

Note. ^a^ EI = energy intake; ^b^ TBW = total body water; ^c^ SDMT = Symbol Digit Modalities Test—values indicate deviation from the normative MS values, with lower scores representing degree of impairment; ^d^ CVLT-II = California Verbal Learning Test II—values indicate deviation from the normative MS values, with lower scores representing degree of impairment; ^e^ BVMT-R = Brief Visuospatial Memory Test—values indicate deviation from the normative MS values, with lower scores representing degree of impairment; ^f^ LPA = light physical activity; ^g^ MVPA = moderate-to-vigorous physical activity.

**Table 3 nutrients-17-00438-t003:** Pearson’s correlations among differences in energy intake and hydration, cognition, and physical activity among a sample of persons with multiple sclerosis in Birmingham, AL, N = 28.

Measure	Total Energy Expenditure	EI ^a^	Absolute Difference EI	Percent Difference EI	TBW ^b^	TBW %	SDMT ^c^	CVLT-II ^d^	BVMT-R ^e^	LPA ^f^
**Total Energy Expenditure**	-									
**EI**	0.16	-								
**Absolute** **Difference EI**	0.43 *	−0.60 ***	-							
**Percent** **Difference EI**	0.40 *	0.72 ***	0.97 ***	-						
**TBW**	0.41 *	−0.08	0.22	0.32	-					
**TBW%**	0.15	0.08	0.06	0.02	−0.11	-				
**SDMT *z*-score**	0.09	0.15	−0.37	−0.35	−0.12	−0.19	-			
**CVLT-II *z*-score**	−0.35	0.10	−0.53 **	−0.48 *	−0.12	−0.06	0.57 ***	-		
**BVMT-R** ***z*-score**	0.13	0.01	−0.23	−0.18	0.10	0.08	0.61 ***	0.58 ***	-	
**LPA minutes/day**	0.33	−0.10	0.46 *	0.42 *	0.08	0.03	−0.13	−0.18	−0.04	-
**MVPA ^g^ minutes/day**	0.21	0.23	−0.05	−0.09	−0.12	0.08	0.20	0.32	0.20	0.38 *

^a^ EI = energy intake; ^b^ TBW = total body water; ^c^ SDMT = Symbol Digit Modalities Test; ^d^ CVLT-II = California Verbal Learning Test II; ^e^ BVMT-R = Brief Visuospatial Memory Test Revised; ^f^ LPA = light physical activity; ^g^ MVPA = moderate-to-vigorous physical activity; * *p* < 0.05, ** *p* < 0.01, *** *p* < 0.001.

**Table 4 nutrients-17-00438-t004:** Linear regression analyses with simultaneous entry of variables for identifying independent contribution of variables explaining differences in energy intake among a sample of 28 persons with multiple sclerosis.

Variable, Units	B	95% CI	β	*T*	*p*
**Absolute Difference EI ^a^**					
Constant CVLT-II ^b^ z-score LPA ^c^ minutes/day Note: *R*^2^ = 0.40	−363.94 −233.39 2.44	−1019.22, 291.34 −402.33, −64.46 0.25, 4.63	- −0.45 0.36	−1.14 −2.85 2.30	**0.26** **0.01** **0.03**
**Percent Difference EI,** %					
Constant CVLT-II z-score LPA minutes/day Note: *R*^2^ = 0.33	−9.90 −8.54 0.09	−37.90, 18.11 −15.76, −1.32 −0.01, 0.18	- −0.41 0.33	−0.73 −2.44 1.95	0.47 **0.02** 0.06

Note. ^a^ EI = energy intake; ^b^ CVLT-II = California Verbal Learning Test II; ^c^ LPA = light physical activity.

## Data Availability

Data described in the manuscript, the code book, and the analytic code will not be made available because the participants did not consent to data availability.

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
