# Peer review of "Correlates of Inaccuracy in Reporting of Energy Intake Among Persons with Multiple Sclerosis"

_nutrients, 2025, doi:10.3390/nu17030438_

Round 1
Reviewer 1 Report
Comments and Suggestions for Authors
Review for the manuscript ,,Correlates of inaccuracy in reporting of energy intake among persons with multiple sclerosis,,
The study want to analyze the the role of the potential variables that contribute to the inter-individual variation in accuracy of reporting of energy intake in a sample of persons with MS. These variables include cognition, hydration status and physical activity.
The results indicated that verbal learning, memory and lower intensity physical activity were associated with inaccuracy in reporting of energy intake in persons with MS.
The manuscript provide contribution for the management of patients with multiple sclerosis and also the results helps clinicians to manage the factors that could improve the quality of life in this medical condition.
Very good description of the methods used. The method provides a lot of details, so that the study protocol can be reproduced in other studies on other populations.
A very good presentation of the results in tables.
Author Response
Comment 1: The study want to analyze the the role of the potential variables that contribute to the inter-individual variation in accuracy of reporting of energy intake in a sample of persons with MS. These variables include cognition, hydration status and physical activity. The results indicated that verbal learning, memory and lower intensity physical activity were associated with inaccuracy in reporting of energy intake in persons with MS. The manuscript provide contribution for the management of patients with multiple sclerosis and also the results helps clinicians to manage the factors that could improve the quality of life in this medical condition. Very good description of the methods used. The method provides a lot of details, so that the study protocol can be reproduced in other studies on other populations. A very good presentation of the results in tables.
Author Response: Thank you for your time reviewing our manuscript and your positive evaluation of the Methods and Results.
Reviewer 2 Report
Comments and Suggestions for Authors
The authors describe their work on potential variables that correlate with inaccuracy of self-reported energy intake (EI) in adults with multiple sclerosis (MS). It was found that California Verbal Learning Test—Second Edition (CVLT-II) z-scores and light physical activity (LPA) were significantly associated with mean absolute difference in EI. CVLT-II z-scores and LPA were the only variables significantly associated with mean percent difference in EI. The regression analyses indicated that both CVLT-II and LPA significantly explained variance in mean absolute difference in EI, and only CVLT-II explained variance for percent difference in EI. The authors concluded that verbal learning and memory and LPA are associated with inaccuracy of self-reported EI in adults with MS. This may guide timely research identifying appropriate protocols for assessment of diet in MS. This is an interesting study. Appropriate methodology has been employed and the conclusions appear to be justified based on the data at hand. I have a few recommendations for consideration.
1. Introduction. It is not clear as to what the determined issue is and its clinical relevance. It is suggested that the authors provide a stronger rationale for the study and a clear hypothesis to be tested in the study.
2. Introduction. Ref [13]. The study by Portaccio and Amato; PMID: 39483763 may be a better citation.
3. Methods. It is interesting to note that 24/28 study participants are female and thus extrapolation to the general MS population may be biased to women with MS.
4. Results/Discussion. It would be interesting to explore if there are any differences due to ethnicity?
5. Discussion. The authors need to emphasize and elaborate on the novelty aspect of their work.
6. Discussion. Please expand on the clinical significance/applicability/solution of the findings.
Author Response
Thank you for your time reviewing our manuscript and overall positive evaluation. The manuscript is now revised based on your comments below, and we believe the revisions improved the overall quality of the paper.
Comment 1: Introduction. It is not clear as to what the determined issue is and its clinical relevance. It is suggested that the authors provide a stronger rationale for the study and a clear hypothesis to be tested in the study.
Response 1: Thank you, we now provide clear rationale and clinical relevance for the current study (L93-94) and highlight the hypothesis presented at the end of the Introduction (L98-100).
Comment 2: Introduction. Ref [13]. The study by Portaccio and Amato; PMID: 39483763 may be a better citation.
Response 2: Thank you, Reference #13 is now updated to align with your recommendation.
Comment 3: Methods. It is interesting to note that 24/28 study participants are female and thus extrapolation to the general MS population may be biased to women with MS.
Response 3: This is a good point. The demographics in this study align with the general MS population (75-80% female); however, we thought it was an important point to highlight in the limitations that is now included (L356-358).
Comment 4: Results/Discussion. It would be interesting to explore if there are any differences due to ethnicity?
Response 4: Thank you, we now included this as part of the Discussion (L360-362). We are hopeful larger, ongoing cohort studies can address this concern.
Comment 5: Discussion. The authors need to emphasize and elaborate on the novelty aspect of their work.
Response 5: Thank you, this is a great point. We highlight L294 and L369, where we highlight this as the first study of MS-specific factors associated with inaccuracy of self-report energy intake and the novelty of this work. We agree that this could be elaborated upon-- we added an additional sentence in the Discussion to highlight this point (L296-298)
Comment 6: Discussion. Please expand on the clinical significance/applicability/solution of the findings.
Response 6: Thank you, we have included a statement regarding the clinical significance in the Conclusions (L377-379).